# Perceptual-Distortion Balanced Image Super-Resolution is a Multi-Objective Optimization Problem

Qiwen Zhu
State Key Lab of MIIPT, Huazhong
University of Science and Technology
Wuhan, China
zhuqiwen@hust.edu.cn

Yanjie Wang
School of AIA, Huazhong University
of Science and Technology
Wuhan, China
aiawyj@hust.edu.cn

Shilv Cai
School of AIA, Huazhong University
of Science and Technology
Wuhan, China
caishilv@hust.edu.cn

Liqun Chen
School of AIA, Huazhong University
of Science and Technology
Wuhan, China
chenliqun@hust.edu.cn

Jiahuan Zhou
Wangxuan Institute of Computer
Technology, Peking University
Beijing, China
jiahuanzhou@pku.edu.cn

Luxin Yan
State Key Lab of MIIPT, Huazhong
University of Science and Technology
Wuhan, China
yanluxin@hust.edu.cn

Sheng Zhong
State Key Lab of MIIPT, Huazhong
University of Science and Technology
Wuhan, China
zhongsheng@hust.edu.cn

Xu Zou[*]
State Key Lab of MIIPT, Huazhong
University of Science and Technology
Wuhan, China
zoux@hust.edu.cn

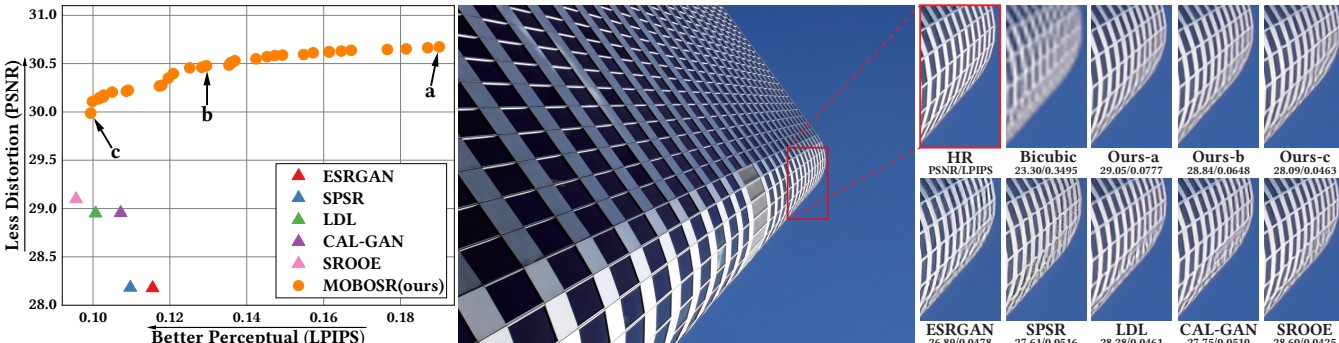

**Figure 1: Left:** Comparisons on the DIV2K [1] validation set, our method not only significantly surpasses others in distortion metrics (PSNR) but also holds an advantage in perceptual quality (LPIPS [58]). [a, b, c] represent three different sample points on the perception-distortion Pareto frontier to illustrate the trade-off between distortion and perceptual quality. **Right:** From the quantitative and visual comparison of this single image, it is evident from our results, labeled as Ours-[a, b, c], that as PSNR decreases and LPIPS increases along the Pareto frontier, the images become sharper. However, this sharpness is accompanied by the introduction of noise and artifacts. In contrast to other methods, our method generates fewer artifacts and avoids excessive blurring, which can be attributed to our model achieving an optimal balance between perceptual quality and distortion.

[*]Corresponding author.

## Abstract

In this paper, we introduce a novel approach to single-image super-resolution (SISR) that balances perceptual quality and distortion through multi-objective optimization (MOO). Traditional pixel-based distortion metrics like PSNR and SSIM often fail to align with human perceptual quality, resulting in blurry outputs despite high scores. To address this, we propose the Multi-Objective Bayesian Optimization Super-Resolution (MOBOSR) framework, which dynamically adjusts loss weights during training. This reduces the need for manual hyperparameter tuning and lessens computational demands compared to AutoML. Our method conceptualizes the relationship between loss weights and image quality assessment (IQA)

metrics as black-box objective functions, optimized to achieve an optimal perception-distortion Pareto frontier. Extensive experiments demonstrate that MOBOSR surpasses current state-of-the-art methods in both perception and distortion, significantly advancing the perception-distortion Pareto frontier. Our work lays a foundation for future exploration of the balance between perceptual quality and fidelity in image restoration tasks. Source codes and pretrained models are available at: https://github.com/ZhuKeven/MOBOSR.

## CCS Concepts

• **Computing methodologies → Reconstruction**.

## Keywords

Super Resolution, Perceptual Quality, Distortion, Multi-Objective Optimization, Loss Function

**ACM Reference Format:**
Qiwen Zhu, Yanjie Wang, Shilv Cai, Liqun Chen, Jiahuan Zhou, Luxin Yan, Sheng Zhong, and Xu Zou. 2024. Perceptual-Distortion Balanced Image Super-Resolution is a Multi-Objective Optimization Problem. In *Proceedings of the 32nd ACM International Conference on Multimedia (MM '24), October 28-November 1, 2024, Melbourne, VIC, Australia.* ACM, New York, NY, USA, 10 pages. https://doi.org/10.1145/3664647.3681512

## 1 Introduction

Super-resolution (SR) plays a crucial role in multimedia, increasingly integrated into devices to enhance the viewing experience of low-resolution images and videos, not only improve visual quality but also reducing bandwidth and storage needs for media streaming.

In the single-image super-resolution (SISR) field, both model architecture and computational efficiency have received considerable attention, resulting in impressive achievements. The majority of these studies report and compare results using PSNR and SSIM [55], both of which are pixel-based metrics. Thus, SR task is actually treated as a regression problem, using regression losses to train, such as L1 Normalization (Mean Absolute Error), L2 Normalization (Mean Squared Error), Charbonnier loss [22], and Huber loss [23]. Pixel-based regression losses are favored for their simplicity and computational efficiency. However, as image downscaling results in significant loss of high-frequency details, SR poses an ill-posed problem. Pixel-based regression loss functions struggle with recovering these high-frequency details [26, 40], often producing blurred images, despite achieving high PSNR and SSIM [55] scores.

While PSNR or SSIM [55] are convenient for computation and comparison, they do not adequately reflect the perceptual quality of images and may even have a negative correlation with human perception (see Figure 1, 'Ours-a' has higher PSNR than 'Ours-c' while 'Ours-a' is more blurry on the contrary). The most accurate reflection of human perceptual quality is naturally human-opinion-scores, but due to its irreproducibility and high cost, it is challenging to use for fair comparative analysis [4]. Consequently, Full Reference Image Quality Assessment (FR-IQA) metrics like LPIPS [58] have been developed to better align with human perception while being reproducible and computationally efficient. To enhance the perceptual quality of SR models, researchers commonly employ perceptual loss [19] or GAN [16] for training, which, while enhancing image sharpness, may also result in the introduction of noise

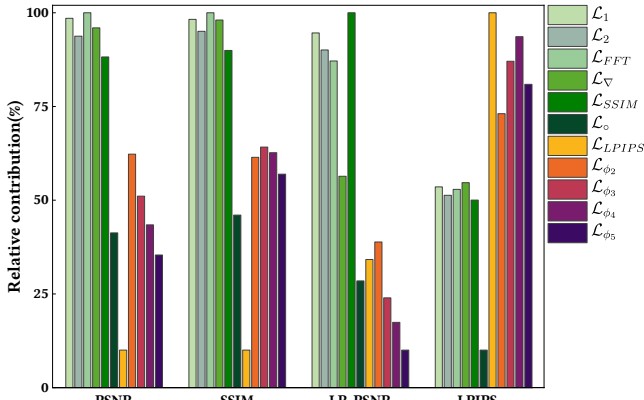

Figure 2: Relative contributions of each loss function to the final model metrics. The definitions of these loss functions are detailed in Section 4.1. The relative contribution are calculated by training a SR model with each individual loss function and evaluating them on the DIV2K [1] validation set. Metrics are normalized, so a higher bar indicates better performance for the corresponding metric. This indicates there is an inherent conflict between perceptual and regression losses (e.g. [$\mathcal{L}_1$, $\mathcal{L}_2$, $\mathcal{L}_{FFT}$, $\mathcal{L}_\nabla$, $\mathcal{L}_{SSIM}$] contribute more to distortion metrics (PSNR, SSIM [55], and LR-PSNR), while [$\mathcal{L}_{LPIPS}$, $\mathcal{L}_{\phi_2}$, $\mathcal{L}_{\phi_3}$, $\mathcal{L}_{\phi_4}$, $\mathcal{L}_{\phi_5}$] contribute more to perceptual metric (LPIPS [58])).

and artifacts (see Figure 1, 'SPSR' [31] holds better LPIPS [58] than 'Ours-b', but 'SPSR' [31] causes much more artifacts).

Therefore, many researchers combine regression and perceptual losses to strive for an optimal balance between pixel accuracy and visual perception [10, 14, 15, 26, 28, 37, 40, 47, 50, 53]. However, as shown in Figure 2, there is an inherent conflict between perceptual loss and regression losses, as well as between perceptual quality and distortion metrics [5], leading to a compromise.

Several methods attempts to balance perception and distortion by manually setting the weights of different losses during training [15, 24, 38, 41, 53]. Concurrently, some approaches involve training two structurally identical networks: one focuses on regression loss to optimize distortion metrics, while the other emphasizes perceptual loss to enhance perceptual metrics. The parameters of these networks are subsequently blended through interpolation, employing manually determined weights to finely adjust the trade-off between perception and distortion within the final model [47, 52, 53]. In a novel strategy, Wang et al. [50] incorporated conditional branches into the network, which are regulated by a scalar value to produce varying trade-offs. Furthermore, certain methods selectively apply different losses to different regions or frequency components of an image [10, 14, 26, 28, 37, 40, 51]. This entails utilizing perceptual loss or GAN for areas with intricate textures to maximize perceptual quality, while reserving pixel regression loss for regions with simpler textures to maintain fidelity. While these artworks have garnered notable success, they have not yet explored the **optimal**

**balance boundary** between perception and distortion. Furthermore, they either necessitate laborious hyperparameter tuning or impose additional computational demands during inference phase.

To intuitively illustrate the conflict between perception and distortion, we conducted a series of simple experiments to quantify the contributions of various loss functions to different metrics, as depicted in Figure 2. It was observed that loss functions with a higher contribution to distortion metrics inversely contributed less to perceptual metrics, and vice versa. This conflicting characteristic prompted us to consider: **Whether Multi-Objective Optimization (MOO) could be a viable solution to this dilemma?**

MOO is an optimization strategy designed to simultaneously address multiple objectives. Due to potential contradictions among these objectives, it is often challenging to find a single solution that maximizes or minimizes all objectives concurrently. Therefore, the aim of MOO is to identify a set of solutions that offer the best compromise among the different objectives (or the so-called optimal balance boundary), culminating in what is known as the Pareto frontier. On this frontier, any improvement in one objective comes at the cost of compromising at least one other objective. Multi-Objective Bayesian Optimization (MOBO), a widely utilized algorithm within the MOO framework, is particularly adept at optimizing black-box objective functions that are gradient-inaccessible and costly to evaluate.

This paper introduces a novel application of MOBO to address the challenge of balancing perceptual quality and distortion in SISR models. We conceptualize the relationship between loss weights and IQA metrics as black-box objective functions to be optimized. In the training process, the performance metrics of SR models guide the optimization algorithm to refine the loss weights for the next epoch, thereby creating a feedback loop that enhances model efficacy with each iteration. This approach circumvents the computational burden associated with AutoML, where each optimization iteration requires a full model training, and does not incur any additional computational or storage costs during inference.

In summary, the contributions of this work are threefold:

- We pioneer the application of MOBO to address the challenge of balancing perception quality and distortion in SISR models.
- We have developed a method for dynamically adjusting loss weights during model training, thereby obviating the need for hyperparameter tuning and significantly reducing the computational resources required in comparison to AutoML hyperparameter search methods.
- This research expands the perceptual-distortion Pareto frontier of the SISR domain, offering new insights into the trade-offs between perceptual quality and distortion in various image restoration tasks.

## 2 Related Works

Our work employs MOBO to address the balance between perception and distortion in SISR models. The following sections will provide an overview of SISR methods aimed at addressing this balance, followed by a concise introduction to MOO methods.

### 2.1 Single Image Super Resolution

Since Dong et al. [11] first applied CNN to the task of SISR, this field has been predominantly driven by deep learning advancements. SISR models have continuously improved in performance. Notably, ESPCN [42] has replaced interpolation with sub-pixel convolution and deferred scaling to the end of the model, significantly reducing computational demands. EDSR [27] has achieved enhanced performance by omitting Batch Normalization [18] layers, which, despite their effectiveness in other tasks, proved detrimental to SR tasks. Furthermore, SwinIR [25] has constructed SR models based on Swin Transformer [29], establishing a new SOTA. These methods employ regression loss for training, achieving high PSNR and SSIM [55] scores but failing to recover high-frequency details, resulting in blurred images.

Johnson et al. [19] introduced perceptual loss which utilizes feature maps extracted from a pre-trained deep convolutional network (such as VGG [43]) as the input for the loss function, significantly enhancing the perceptual quality of super-resolution models. SR-GAN [24] pioneered the application of GAN [16] to SR models, enabling the generation of sharp details. These two seminal works have inspired a plethora of perception-oriented SR approaches. Beyond perceptual [19] and GAN [16] losses, a plethora of other loss functions have been incorporated into SISR models. A category of these introduces additional constraints, including: frequency domain constraints [6, 10, 15, 28, 44, 49], which utilize the spectral maps generated by FFT, DCT or Wavelet transformations to compute loss; gradient constraints [6, 30, 31, 46, 47, 54], which calculate loss using the gradient maps of images; and low-resolution consistency constraints [34, 46], which compute loss by comparing the downsampled super-resolved images with the original low-resolution images, thereby reducing the likelihood of generating incorrect textures. The application of these losses can further enhance certain metrics of SISR models, as shown in Figure 2.

### 2.2 Perceptual Distortion Trade-Off

However, while perceptual loss [19] and GAN [16] can enhance perceptual quality, they also introduce noise or artifacts and may even produce incorrect textures. To balance perceptual quality and distortion in SISR, most methods involve manually adjust the weights of different loss functions [15, 24, 38, 41, 53], necessitating extensive hyperparameter tuning to achieve superior outcomes, yet without guaranteeing optimality. Vu et al. [47], along with Wang et al. [52, 53], have demonstrated that a balance between perception quality and distortion can be attained through network parameter interpolation. This method mitigates the need for extensive network training for each weight by limiting the process to just two networks, thereby substantially reducing resource consumption. However, it is important to note that this approach still does not guarantee optimal results. In a similar way, Wang et al. [50] introduced conditional branches within the network architecture, modulated by a scalar parameter, to facilitate a spectrum of trade-offs. However, this approach introduces additional computational load during inference phase.

Rad et al. [40] categorized images into background, edges, and objects, applying different loss functions to each category. Fritsche et al. [14] employed separate loss functions for the high-frequency

and low-frequency components of images. Liang et al. [26] divided images into numerous patches, classifying them into three categories—solid colors, texture-rich like trees or hair, and those with strong edge transitions—and applied different loss functions accordingly. Liu et al. [28] segmented images based on Spectral Bayesian Uncertainty before applying various loss functions. Park et al. [37] train an auxiliary network using the LPIPS [58] Map as supervisory signal to predict a discrete set of predefined loss weight combinations, aiming to maximum perceptual quality. These methods either fail to achieve the optimal balance, necessitate additional network training, or increase computational load during inference. Decomposing images before super-resolution and then recombining them may also potentially introduce more noise or artifacts.

## 2.3 Multi-Objective Optimization

MOO is a process that seeks to find the optimal balance among several often conflicting objectives. If the user has a clear preference for each objective, a weighted sum approach can be employed to transform it into a single-objective optimization problem. Conversely, if the user's preferences are not well-defined, MOO techniques are necessary to determine the Pareto frontier.

In the field of MOO, the two main methodologies are multi-objective evolutionary algorithms (MOEA) and multi-objective Bayesian optimization (MOBO). MOEAs, such as NSGA-II [9], SPEA2 [59], MOEA/D [57], SMPSO [35] etc., are well-regarded for their ability to generate a diverse set of Pareto-optimal solutions through population-based approaches. They excel in exploring the solution space, which is crucial for capturing the trade-offs among conflicting objectives. However, MOEAs can be computationally demanding, especially when scaling to problems with high-dimensional objective spaces or when requiring a high-resolution Pareto front.

MOBO approaches extends Bayesian Global Optimization (BGO) [20] from single-objective to multi-objective contexts. It typically uses Gaussian Processes (GP) for modeling the objective functions and utilizes heuristic acquisition functions, such as Expected Hypervolume Improvement (EHVI) [7, 8, 13, 48], to identify the subsequent data point for evaluation. MOBO is highly sample efficient and is particularly well-suited for optimizing black-box objective functions which are expensive to evaluate. Thus, MOBO is especially well-suited for implementing our proposed SISR approach that balances perceptual quality and distortion.

## 3 Method

### 3.1 Preliminary

MOO refers to simultaneously optimizing multiple objective functions $f(x) \subset \mathbb{R}^M$ over a bounded search space $\mathcal{X} \in \mathbb{R}^d$, aiming to find solutions that optimize all objectives. However, these objectives often conflict with each other, making it challenging to find a single solution that optimizes all objectives simultaneously. Instead, we typically obtain a compromise among multiple objectives. A common approach is to identify the **Pareto front**.

> **Definition 1:** Given $f(x) = [f_1(x), \ldots, f_M(x)]$ and $f(x') = [f_1(x'), \ldots, f_M(x')]$, if $f_i(x) \geq f_i(x')$ for all $i = 1, \ldots, M$, and there exists at least one $i \in 1, \ldots, M$ such that $f_i(x) > f_i(x')$, we say that $f(x)$ **dominates** $f(x')$, denoted as $f(x) > f(x')$.

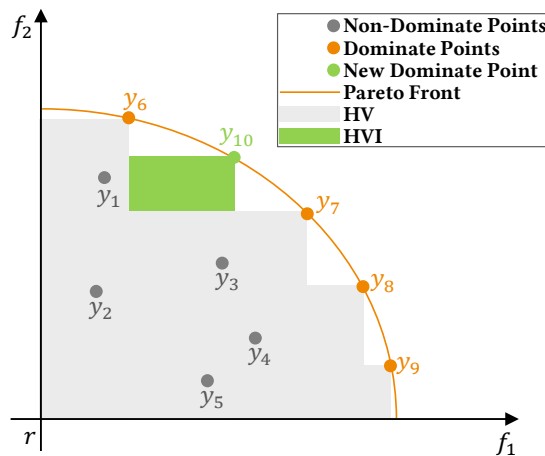

**Figure 3: A toy example that demonstrates the 2-dimensional Pareto frontier, along with the HV and HVI.**

> **Definition 2:** An $f(x^*)$ is considered **Pareto optimal** if no other solution can dominate it.
>
> **Definition 3:** In the space of objective functions, the set of all Pareto optimal solutions forms the **Pareto front**: $\mathcal{P} = \{f(x) \ s.t. \ \nexists x' \in \mathcal{X} : f(x') > f(x)\}$.
>
> **Definition 4: Hypervolume (HV)** quantifies the volume of the space dominated by a finite approximate Pareto front $\mathcal{P}$, with respect to a reference point $r \in \mathbb{R}^M$. Specifically, it measures the M-dimensional Lebesgue measure $\lambda_M$ of the region that is bounded by the hyper-rectangle formed by vertices $r$ and $y_i$: $HV(\mathcal{P}, r) = \lambda_M(\cup_{i=1}^{|\mathcal{P}|} [r, y_i])$.
>
> **Definition 5: Hypervolume Improvement (HVI)** assesses the improvement achieved by a set of points $\mathcal{Y}$ concerning a given Pareto front $\mathcal{P}$, relative to a reference point $r$: $HVI(\mathcal{Y}, \mathcal{P}, r) = HV(\mathcal{P} \cup \mathcal{Y}, r) - HV(\mathcal{P}, r)$.

Figure 3 presents an intuitive 2-dimensional toy example of these definitions. When optimizing an objective function $f(x)$ that is expensive to evaluate and lacks gradient information, MOBO emerges as the most suitable approach. It constructs a probabilistic surrogate model representing the objective function and employs an acquisition function to determine the next sampling point. Through iterative updates of the surrogate model, MOBO can efficiently identify the global optimum with fewer evaluations.

### 3.2 Problem Formulation

The primary goal of SISR is to employ a model, *e.g.* a neural network, denoted as $G_\theta$ and parameterized by $\theta$, to transform a low-resolution input image $I^{LR}$, which is typically obtained through bicubic downsampling from a high-resolution image $I^{HR}$, into a super-resolved image $I^{SR}$ with the same resolution as $I^{HR}$:

$$I^{SR} = G_\theta(I^{LR}). \tag{1}$$

The goal is to make $I^{SR}$ as close as possible to $I^{HR}$, and to achieve this, we minimize a loss function:

$$\arg\min_\theta \mathcal{L}(G_\theta(I^{LR}), I^{HR}). \tag{2}$$

However, relying solely on a single loss function may not lead to an optimal model, as different loss functions have different advantages and drawbacks. Therefore, we linearly combine multiple losses:

$$\mathcal{L} = \sum_{i=1}^{N} \omega_i \mathcal{L}_i. \tag{3}$$

The gradient of this combined loss with respect to $\theta$ is:

$$\frac{\partial \mathcal{L}}{\partial \theta} = \sum_{i=1}^{N} \omega_i \frac{\partial \mathcal{L}_i}{\partial \theta}. \tag{4}$$

Therefore, by adjusting the weights $\omega_i$ of each loss $\mathcal{L}_i$, we can control the corresponding magnitude of the loss gradient and, consequently, its impact on the IQA metrics $Q_\theta = [PSNR_\theta, \ldots, LPIPS_\theta]$ of SR model. There exists a deterministic relationship between $Q_\theta$ and loss weights $\omega = [\omega_1, \ldots, \omega_N]$:

$$Q_\theta = f(\omega). \tag{5}$$

Solving for $f$ allows us to achieve optimal balance between perceptual quality and distortion by setting the best loss weights. However, since $f$ is an unknown black-box function and these objectives may conflict with each other, MOBO will be employed for its estimation.

## 3.3 Multi-Objective Bayesian Optimization Super-Resolution

Initially, we randomly and uniformly sample a certain number of initial weights within the search space $\omega \in \Omega^d$, train the SR model using the sampled weights, and then compute the IQA metrics of the SR model to obtain the observed dataset $\mathcal{D} = \{(Q_i, \omega_i)\}_{i=1}^{T}$.

Subsequently, we use the observed data $\mathcal{D}$ to fit a probabilistic surrogate model. Surrogate models typically employ Gaussian processes (GP), which simulates the probability distribution of the true objective function $f$ based on the observed data $\mathcal{D}$:

$$f \sim \mathcal{GP}(\mu, K), \tag{6}$$

where $\mu$ is the mean function, and $K$ is the covariance function. In MOBO, a GP is fitted for each objective.

Given the high cost of evaluating the objective function and the low cost of evaluating the surrogate model, BO utilizes acquisition functions to heuristically select the next data point for evaluating and training, thereby minimizing the number of evaluations. Acquisition functions assess the potential performance of new data point based on the distribution predicted by the surrogate model. The most commonly used acquisition function is the Expected Improvement (EI):

$$\alpha_{EI}(\omega) = \mathbb{E}[\max(0, f(\omega^*) - f(\omega))], \tag{7}$$

which can achieves a good balance between exploration and exploitation. When extended to multi-objective scenarios, the Expected Hypervolume Improvement (EHVI) is used as the acquisition function:

$$\alpha_{EHVI}(\omega|\mathcal{P}) = \mathbb{E}[HVI(f(\omega)|\mathcal{P})]. \tag{8}$$

Through the acquisition function, we iteratively select the next most valuable point for evaluation and update the surrogate model:

$$\omega^* = \arg\max_{\omega \in \Omega^d} \alpha(\omega). \tag{9}$$

---

**Algorithm 1:** MOBOSR Pseudocode

---

**Data:** Pretrain epochs $T_{init}$, total epochs $T$, $i \leftarrow 1$.
**Result:** SR models $\mathcal{M} = \{M_1, \ldots, M_N\}$ on optimal distortion and perceptual Pareto frontier.

1   define optimization objectives $Q$, parameters $\omega$, dataset $\mathcal{D}$, objective function $f \sim \mathcal{GP}(\mu, K)$, acquisition function $\alpha_{EHVI}(\omega|\mathcal{P})$, SR model $M$;

2   **for** $i = 1$ *to* $T_{init}$ **do**

3      random sample $\omega_i \in \Omega^d$;

4      $M$.train($\omega_i$);

5      $Q_i = M$.eval();

6      $\mathcal{D} = \mathcal{D} \cup \{(Q_i, \omega_i)\}$;

7   **end**

8   **for** $i = 1$ *to* $T$ **do**

9      fit $f \sim \mathcal{GP}(\mu, K)$ using $\mathcal{D}$;

10     $\omega_i = \arg\max_{\omega \in \Omega^d} \alpha_{EHVI}(\omega|\mathcal{P})$;

11     $M$.train($\omega_i$);

12     $Q_i = M$.eval();

13     $\mathcal{D} = \mathcal{D} \cup \{(Q_i, \omega_i)\}$;

14   **end**

15   obtain $\mathcal{M}$ through compute Pareto frontier from $\mathcal{D}$;

---

By repeating these steps, BO can achieve the global optimum of the black-box objective function with a minimal number of evaluation steps. The pseudocode for the overall optimization process is outlined as Algorithm 1.

# 4 Experiments

## 4.1 Implementation Details

*Model Architecture.* The ESRGAN [53] architecture was employed as the SR model to evaluate the approach, due to its excellent performance and popularity in recent perception-related SR research. The widespread adoption of ESRGAN [53] and its proposed RRDB [53] backbone ensures a comprehensive and equitable comparison of our methodology.

*Loss Functions.* The loss functions we employed include:

- L1 Loss:

$$\mathcal{L}_1 = \left\| I^{SR} - I^{HR} \right\|_1. \tag{10}$$

- L2 Loss:

$$\mathcal{L}_2 = \left\| I^{SR} - I^{HR} \right\|_2^2. \tag{11}$$

- FFT Loss:

$$\mathcal{L}_{FFT} = \left\| FFT(I^{SR}) - FFT(I^{HR}) \right\|_1. \tag{12}$$

- Gradient Loss:

$$\mathcal{L}_\nabla = \left\| \nabla_h I^{SR} - \nabla_h I^{HR} \right\|_1 + \left\| \nabla_v I^{SR} - \nabla_v I^{HR} \right\|_1, \tag{13}$$

  where $\nabla_h$ and $\nabla_v$ are the Sobel gradient operators in the horizontal and vertical directions.

- Perceptual Loss [19]:

$$\mathcal{L}_{\phi_j} = \left\| \phi_j(I^{SR}) - \phi_j(I^{HR}) \right\|_1, j \in \{2, 3, 4, 5\}, \tag{14}$$

  Qiwen Zhu, Yanjie Wang, Shilv Cai, Liqun Chen, Jiahuan Zhou, Luxin Yan, Sheng Zhong, and Xu Zou

**Table 1: Comparison of MOBOSR with other artworks on 7 datasets. The best and second-best results are highlighted in bold and underline, respectively. The symbols ↑ and ↓ indicate that higher or lower values of the metric are preferable. To the best of our knowledge, for fair comparisons, all publicly available methods (utilize RRDB [53] as the backbone, and aim to address the balance between perceptual quality and distortion) are selected. This includes SPSR [31], RRDB+LDL [26], CAL-GAN [36] and SROOE [37]. In this table, we use the data point labeled as Our-c in Figure 1 for comparison. Except for SROOE [37], our proposed MOBOSR consistently outperforms existing methods by a large margin on all metrics, particularly demonstrating a significant advantage in LR-PSNR with over 5dB improvements. Compared to SROOE [37], we achieve superior performance on all distortion metrics with similar LPIPS [58], even though we only train on the smaller DIV2K [1] training set. Quantitative comparisons of Ours-[a,b] and details on metric evaluation are available in the supplementary material.**

| Metrics | Methods | Train Datasets | Set5 | Set14 | DIV2K | BSD100 | Urban100 | General100 | Manga109 |
|---|---|---|---|---|---|---|---|---|---|
| PSNR↑ | ESRGAN [53] | DF2K-OST | 30.4618 | 26.2839 | 28.1778 | 25.2892 | 24.3617 | 29.4593 | 28.5041 |
| | SPSR [31] | DIV2K | 30.3871 | 26.6501 | 28.1824 | 25.4949 | 24.8063 | 29.4794 | 28.6102 |
| | RRDB+LDL [26] | DIV2K | 31.0007 | 27.2064 | 28.9510 | 26.0988 | 25.4781 | 30.1974 | 29.4111 |
| | CAL-GAN [36] | DIV2K | 31.0475 | 27.3272 | 28.9549 | 26.2581 | 25.2908 | 30.0742 | 29.1665 |
| | SROOE [37] | DF2K | 31.2455 | 27.2561 | 29.0990 | 26.1715 | 25.8452 | 30.4723 | 29.9017 |
| | Ours-c | DIV2K | **31.8272** | **28.1766** | **29.9858** | **27.0494** | **26.0764** | **31.1164** | **30.2763** |
| SSIM↑ | ESRGAN [53] | DF2K-OST | 0.8518 | 0.6982 | 0.7761 | 0.6496 | 0.7341 | 0.8102 | 0.8604 |
| | SPSR [31] | DIV2K | 0.8432 | 0.7133 | 0.7720 | 0.6571 | 0.7472 | 0.8095 | 0.8591 |
| | RRDB+LDL [26] | DIV2K | 0.8610 | 0.7343 | 0.7952 | 0.6811 | 0.7670 | 0.8278 | 0.8746 |
| | CAL-GAN [36] | DIV2K | 0.8552 | 0.7353 | 0.7897 | 0.6789 | 0.7623 | 0.8262 | 0.8676 |
| | SROOE [37] | DF2K | 0.8651 | 0.7304 | 0.7980 | 0.6866 | 0.7764 | 0.8332 | 0.8786 |
| | Ours-c | DIV2K | **0.8804** | **0.7615** | **0.8203** | **0.7109** | **0.7812** | **0.8495** | **0.8938** |
| LR-PSNR↑ | ESRGAN [53] | DF2K-OST | 46.7348 | 43.8433 | 45.9012 | 43.8190 | 42.9339 | 45.4220 | 43.9667 |
| | SPSR [31] | DIV2K | 46.3607 | 43.6201 | 44.8529 | 42.6756 | 42.6679 | 44.6786 | 44.3872 |
| | RRDB+LDL [26] | DIV2K | 48.5067 | 46.2893 | 47.9757 | 45.1571 | 46.5827 | 48.0079 | 47.8923 |
| | CAL-GAN [36] | DIV2K | 42.4327 | 41.5963 | 42.8611 | 41.0666 | 41.6069 | 43.4227 | 42.8636 |
| | SROOE [37] | DF2K | 53.1781 | 51.0679 | 53.5488 | 51.2347 | 50.6700 | 52.9797 | 51.7820 |
| | Ours-c | DIV2K | **54.3372** | **53.3344** | **55.2161** | **53.3618** | **52.9401** | **54.5283** | **53.4195** |
| LPIPS↓ | ESRGAN [53] | DF2K-OST | 0.0750 | 0.1341 | 0.1155 | 0.1617 | 0.1228 | 0.0876 | 0.0647 |
| | SPSR [31] | DIV2K | 0.0616 | 0.1313 | 0.1097 | 0.1629 | 0.1186 | 0.0866 | 0.0662 |
| | RRDB+LDL [26] | DIV2K | 0.0637 | 0.1309 | 0.1007 | 0.1635 | 0.1097 | 0.0794 | 0.0546 |
| | CAL-GAN [36] | DIV2K | 0.0687 | 0.1320 | 0.1072 | 0.1696 | 0.1171 | 0.0894 | 0.0688 |
| | SROOE [37] | DF2K | **0.0603** | **0.1131** | **0.0956** | 0.1514 | **0.1067** | **0.0758** | **0.0511** |
| | Ours-c | DIV2K | 0.0607 | 0.1240 | 0.0994 | **0.1508** | 0.1154 | 0.0776 | 0.0576 |

where $\phi_j$ is the feature map output by the VGG [43] network's $j$th layer.

- Cycle Consistency Loss:

$$\mathcal{L}_\circ = \left\| \Downarrow^3 (I^{SR}) - I^{LR} \right\|_1, \qquad (15)$$

where $\Downarrow^3$ is the bicubic downsampling operator.

- GAN Loss [16]:
  The discriminator loss is:

$$\mathcal{L}_D = -log(D(I^{SR})), \qquad (16)$$

where $D$ represents the discriminator network. The generator loss is:

$$\mathcal{L}_{GAN} = -log(D(I^{HR})) - log(1 - D(I^{SR})). \qquad (17)$$

- SSIM [55] Loss:

$$\mathcal{L}_{SSIM} = 1 - \frac{(2\mu_{I^{SR}}\mu_{I^{HR}} + C_1)(2\sigma_{I^{SR}I^{HR}} + C_2)}{(\mu_{I^{SR}}^2 + \mu_{I^{HR}}^2 + C_1)(\sigma_{I^{SR}} + \sigma_{I^{HR}} + C_2)}, \qquad (18)$$

where $\mu_{I^{SR}}$ and $\mu_{I^{HR}}$ are the mean values of $I^{SR}$ and $I^{HR}$, $\sigma_{I^{SR}}$ and $\sigma_{I^{HR}}$ are the standard deviations of $I^{SR}$ and $I^{HR}$, $\sigma_{I^{SR}I^{HR}}$ is the covariance between $I^{SR}$ and $I^{HR}$, $C_1$ and $C_2$ are small constants to stabilize the division.

- LPIPS [58] Loss:

$$\mathcal{L}_{LPIPS} = \sum_{l \in F} w_l \left\| \phi_l(I^{SR}) - \phi_l(I^{HR}) \right\|_2^2, \qquad (19)$$

where $F$ represents the set of intermediate layers in the chosen neural network, $\phi_l$ is the normalized outputs of layer $l$ for images, $\omega_l$ are the official weights determined by Zhang et al. [58].

*Datasets.* The DIV2K [1] training set (800 images) was employed for training. For testing, we utilize a collection of datasets including Set5 [3], Set14 [56], DIV2K [1] validation set, BSD100 [32], Urban100 [17], General100 [12], and Manga109 [33]. We followed the previous work [26, 31, 36, 53] to focus only on ×4 super-resolution to train and evaluate the effectiveness of our SR model. Specifically, we

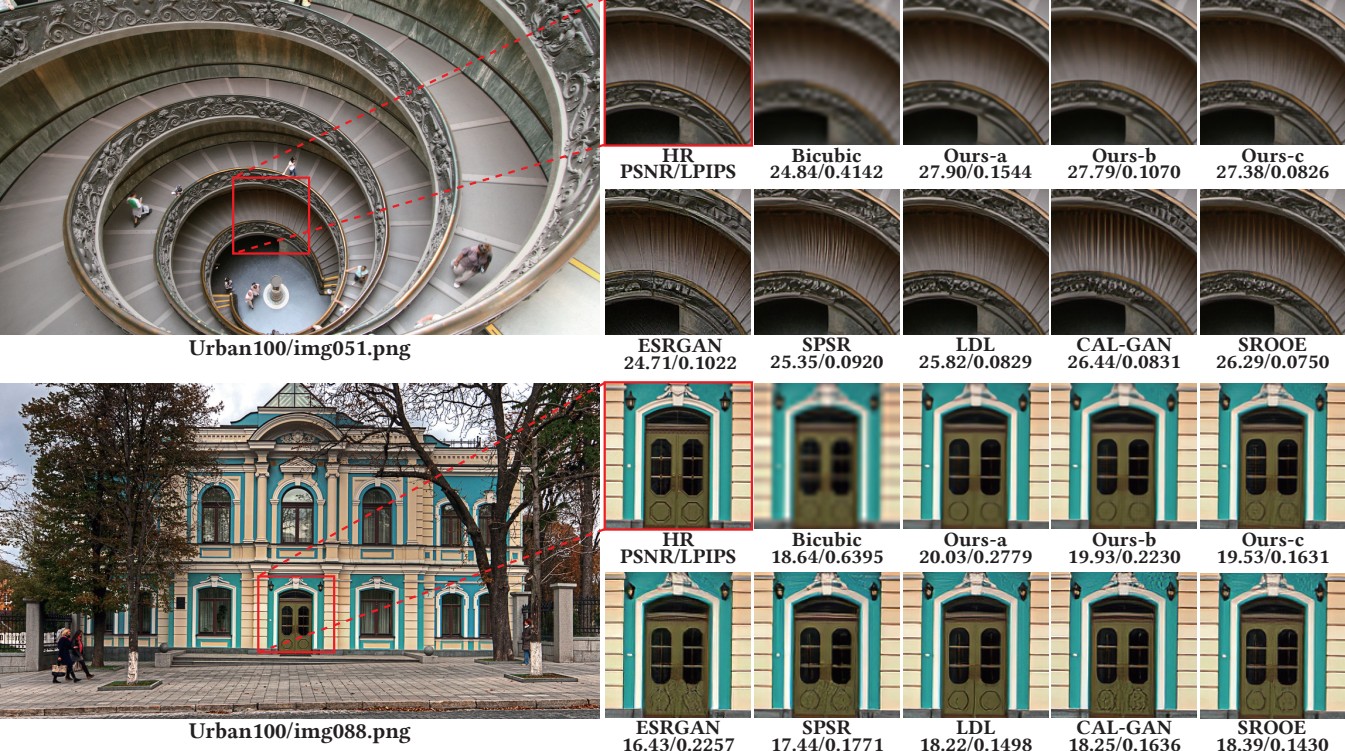

**Figure 4: Visual comparison of three sampled points on the Pareto frontier obtained through our method (as defined in Figure 1), alongside other artworks, on the Urban100 [17] dataset. More visual results are presented in the supplementary material.**

obtain the LR images by downsampling the HR counterparts by a factor of 4 using bicubic interpolation.

*Training Details.* We randomly sample 16 patches from each image as a batch, the patch size of LR is $32 \times 32$. To ensure stability in the SR model training progress and MOBO optimization process, we initially pre-train for 250 epochs using manually set loss weights from ESRGAN [53], defined as:

$$\mathcal{L} = \omega_1 \mathcal{L}_1 + \omega_{\phi_5} \mathcal{L}_{\phi_5} + \omega_{GAN} \mathcal{L}_{GAN}, \quad (20)$$

where $[\omega_1, \omega_{\phi_5}, \omega_{GAN}] = [1e-2, 1, 5e-3]$. Subsequently, we commence loss weight optimization using our MOBO method. The learning rate during pre-training is fixed at $1e-4$, and for optimization with MOBO, the initial learning rate is set at $5e-5$ and is halved every 250 epochs. The optimizer employed is Adam [21]. Our SR model is implemented by PyTorch [39] on NVIDIA RTX 3090 GPUs, and MOBO is realized through the Ax platform [2].

## 4.2 Quantitative Results

Methods utilize RRDB [53] as the backbone, and aim to address the balance between perceptual quality and distortion are selected for fair comparisons. To this end, to the best of our knowledge, all five publicly available artworks are used for comparison: ESRGAN [53], SPSR [31], RRDB+LDL [26], CAL-GAN [36] and SROOE [37]. All of which utilize RRDB [53] as their backbones. It is noteworthy that SROOE [37] is trained on the larger DF2K dataset (3450 images), which consists of the DIV2K [1] training set (800 images)

and Flickr2K [45] (2650 images). ESRGAN [53], on the other hand, is trained on the even larger DF2K-OST dataset (13774 images), which includes DF2K (3450 images) and OST [51] (10,324 images). Additionally, the SPSR [31] network incorporates an extra gradient branch, and SROOE [37] introduces an auxiliary network to predict a discrete set of predefined loss weight combinations. These two methods introduce additional computational load during inference.

We incorporated all the loss functions introduced in Section 4.1 and optimized their weights using the MOBO strategy. Our optimization objectives were PSNR and LPIPS [58]. The loss function is formulated as:

$$\begin{aligned} \mathcal{L} = &\omega_1 \mathcal{L}_1 + \omega_2 \mathcal{L}_2 + \omega_{FFT} \mathcal{L}_{FFT} + \\ &\omega_\nabla \mathcal{L}_\nabla + \omega_{\phi_2} \mathcal{L}_{\phi_2} + \omega_{\phi_3} \mathcal{L}_{\phi_3} + \\ &\omega_{\phi_4} \mathcal{L}_{\phi_4} + \omega_{\phi_5} \mathcal{L}_{\phi_5} + \omega_\circ \mathcal{L}_\circ + \\ &\omega_{GAN} \mathcal{L}_{GAN} + \omega_{SSIM} \mathcal{L}_{SSIM} + \\ &\omega_{LPIPS} \mathcal{L}_{LPIPS}. \end{aligned} \quad (21)$$

And the optimization formula is:

$$[PSNR, LPIPS] = f([\omega_1, \omega_2, \omega_{FFT}, \omega_\nabla, \omega_{\phi_2}, \omega_{\phi_3}, \omega_{\phi_4}, \\ \omega_{\phi_5}, \omega_\circ, \omega_{GAN}, \omega_{SSIM}, \omega_{LPIPS}]). \quad (22)$$

As demonstrated in Table 1, except for SROOE [37], our approach not only demonstrates superior performance in the full-reference perceptual metric (LPIPS [58]) but also significantly surpasses other methods in distortion metrics (PSNR and SSIM [55]) and consistency metric (LR-PSNR). Compared to SROOE [37], we achieve

Qiwen Zhu, Yanjie Wang, Shilv Cai, Liqun Chen, Jiahuan Zhou, Luxin Yan, Sheng Zhong, and Xu Zou

**Table 2: Ablation study on 7 datasets compares MOBO-optimized loss weights to ESRGAN [53] manually-set loss weights. Both models use the official ESRGAN [53] train code and are trained on the DIV2K [1] training set with the same settings, except that the loss weights for MOBOSR are optimized by MOBO. The best results are highlighted in bold. MOBOSR consistently outperforms ESRGAN [53] by a large margin in the same setting. This comparison intuitively demonstrates the effectiveness of the core idea of our work: automatic loss weight optimization.**

| Metrics | Methods | Set5 | Set14 | DIV2K | BSD100 | Urban100 | General100 | Manga109 |
|---------|---------|------|-------|-------|--------|----------|-----------|----------|
| PSNR↑ | ESRGAN [53] | 29.8023 | 25.5164 | 27.6994 | 25.2100 | 23.7431 | 28.8618 | 27.3669 |
| | MOBOSR (Ours) | **30.6926** | **26.8148** | **28.5089** | **26.0402** | **24.4895** | **29.6640** | **28.2091** |
| SSIM↑ | ESRGAN [53] | 0.8456 | 0.6855 | 0.7610 | 0.6463 | 0.7116 | 0.7979 | 0.8436 |
| | MOBOSR (Ours) | **0.8607** | **0.7234** | **0.7834** | **0.6781** | **0.7357** | **0.8209** | **0.8621** |
| LR-PSNR↑ | ESRGAN [53] | 42.5573 | 38.2986 | 41.2244 | 40.3034 | 37.6727 | 41.6356 | 40.0483 |
| | MOBOSR (Ours) | **44.6401** | **43.0412** | **44.6847** | **43.5394** | **41.8108** | **44.6812** | **43.1860** |
| LPIPS↓ | ESRGAN [53] | **0.0742** | 0.1689 | 0.1193 | 0.1751 | 0.1379 | 0.0943 | 0.0748 |
| | MOBOSR (Ours) | 0.0745 | **0.1359** | **0.1145** | **0.1719** | **0.1324** | **0.0894** | **0.0675** |

superior performance on all distortion metrics with similar LPIPS [58], even though we only train on the smaller DIV2K [1] training set. This indicates that our approach not only achieves excellent perceptual metrics but also mitigates the artifacts and false textures introduced by perceptual loss [19] and GAN [16], thereby ensuring consistency with the LR source and the pixel accuracy with the HR image. In summary, our method consistently outperforms others in both distortion and perceptual quality, demonstrating that the integration of MOBO facilitates the attainment of the Pareto frontier (optimal balance boundary) for perception quality and distortion.

Furthermore, prior to our work, manually setting such a multitude of loss weights is impractical, which consequently limits the incorporation of additional loss functions into the SISR model. While our method can efficiently and autonomously determine the optimal weights for loss functions, without introducing any additional computations during the inference and significantly extending the training duration (time consuming about SR model and MOBO during training are detailed in the supplementary material).

### 4.3 Qualitative Results

Visual comparisons of Ours-[a,b,c] (as defined in Figure 1), as depicted in Figure 4, clearly reveal the inherent contradiction between perception quality and distortion. This is evidenced by the gradual deterioration in PSNR and the corresponding improvement in LPIPS [58] on the Pareto frontier. A higher PSNR typically results in a more blurred image, albeit with reduced noise and artifacts. Conversely, a better LPIPS [58] score indicates a sharper image, which, however, tends to introduce noise and artifacts. Compared to other methods, our approach generates fewer noise and artifacts without excessively blurring the image. Additionally, it exhibits more accurate reconstruction in certain areas, better preserving the original structure of the objects in image.

### 4.4 Ablation Study

To demonstrate the effectiveness of using MOBO to optimize loss weights compared to manually set loss weights, we conduct a comparative experiment. Our setup is identical to that of ESRGAN [53], with the exception that the loss weights are optimized by MOBO.

Our optimization formula is expressed as:

$$[PSNR, LPIPS] = f([\omega_1, \omega_{\phi_5}, \omega_{GAN}]). \quad (23)$$

The loss weights for ESRGAN [53] are adopted from the weights utilized by the authors in the original publication: $[\omega_1, \omega_{\phi_5}, \omega_{GAN}] = [1e-2, 1, 5e-3]$. As demonstrated in Table 2, our approach significantly surpasses ESRGAN [53] across all metrics, which is retrained on the DIV2K [1] training set using the official released codes. This experiment clearly demonstrates the substantial advantages of integrating MOBO for automatically optimizing loss weights in the SR model over the manual tuning of loss weights.

From the comparison between 'MOBOSR (Ours)' in Table 2 and 'Ours-c' in Table 1, it is evident that incorporating a greater variety of loss functions significantly benefits the SISR model, provided that the weights for these losses are optimally set.

### 5 Conclusion

In this research, we propose the Multi-Objective Bayesian Optimization Super-Resolution (MOBOSR). To effectively address the balance between perceptual quality and distortion, we introduce multi-objective Bayesian optimization into the single-image super-resolution model, dynamically adjusting the weights of various loss functions during training. Our method is rigorously validated through comprehensive experiments, demonstrating its effectiveness and versatility in both distortion and perceptual quality. Given its potential to be applied to nearly all domains of image restoration and enhancement, we believe our method can offer valuable insights and contributions to the community.

### Acknowledgments

This work was supported in part by the Key Laboratory of Smart Earth under Grant KF2023YB01-13, and in part by the National Natural Science Foundation of China (NSFC) under Grant 62176100. The computation is completed in the HPC Platform of Huazhong University of Science and Technology.

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
