# OpenReview forum: "Perceptual-Distortion Balanced Image Super-Resolution is a Multi-Objective Optimization Problem"
_acmmm.org/ACMMM/2024/Conference — MM2024 Oral_

### Official Review · Reviewer_nmcS · 2024-05-07

**Rating:** 4
**Confidence:** 3

**Summary:**

This paper introduced multi-objective optimization to single-image super-resolution to balance perceptual quality and distortion. Specifically, they developed a method for dynamically adjusting loss weights during model training, thereby obviating the need for hyperparameter tuning and significantly reducing the computational resources required in comparison to AutoML hyperparameter search methods. This research expands the perceptual-distortion Pareto frontier of the single-image super-resolution domain, offering new insights into the tradeoffs between perceptual quality and distortion in various image restoration tasks.

**Strengths:**

Super-resolution is a research hotspot in image processing community. This paper investigates an important and interesting problem for super-resolution, that is, how to balance perceptual quality and distortion? The tradeoff problem also exists in other image enhancement and restoration tasks. This work may inspire similar works and facilitate the developments of image enhancement and restoration.
(1)	The motivation of this paper is reasonable and convincing.
(2)	The theoretical analysis and problem formulation of this paper are clear and easy to understand.

**Limitations:**

The detailed limitations of this paper are as follows.
(1) Authors should provide and discuss the further applications of the proposed method. This helps to enlarge the contribution and impact of this paper.
(2) Both the related work and comparison algorithms do not contain work in 2024. Authors should re-survey their academic community to make sure if they missed recent relevant work.
(3) The generalization ability of the proposed method should be tested.

**Suitability:**

3

---

### Official Review · Reviewer_NxRH · 2024-05-18

**Rating:** 3
**Confidence:** 3

**Summary:**

In this manuscript, the authors explore the trade-off between image distortion and perception in reconstructed image. Building upon this, the authors introduce multi-objective optimization (MOO) into the Single Image Super-Resolution (SISR) domain, aiming to reduce the need for manual adjustment of hyperparameters by dynamically adjusting the weights of loss functions during training.

**Strengths:**

1)The article represents the first attempt to apply MOBO to loss function adjustment in the SISR domain, addressing the challenge of trade-off between perceptual quality and distortion.
2)The proposed method of dynamically adjusting the weights of loss functions is significantly more resource-efficient compared to AutoML hyperparameter search methods.
3)The article conducts comprehensive experiments and maintains a smooth logical flow in its narrative.

**Limitations:**

1)Whether through loss function adjustment or other strategies, the concept of perception-distortion trade-off has been attempted in some previous works on image super-resolution. However, the authors did not adequately review and discuss these methods. Some relevant articles include:
i) Park, S. H., Moon, Y. S., & Cho, N. I. (2023). Perception-oriented single image super-resolution using optimal objective estimation. In Proceedings of the IEEE/CVF Conference on Computer Vision and Pattern Recognition (pp. 1725-1735).
ii)Deng, X., Yang, R., Xu, M., & Dragotti, P. L. (2019). Wavelet domain style transfer for an effective perception-distortion tradeoff in single image super-resolution. In Proceedings of the IEEE/CVF international conference on computer vision (pp. 3076-3085).
2)The article "Perception-oriented single image super-resolution using optimal objective estimation" builds its network based on RRDB, similar to this manuscript, and adopts a strategy of dynamically adjusting loss functions to achieve a balance between perception and distortion. The authors need to provide comparisons with this algorithm, including performance analysis, differences in strategy usage, and the advantages of this article.
3)The authors should include discussions on the limitations of this work and future prospects.

**Suitability:**

3

---

### Official Review · Reviewer_RbMV · 2024-05-19

**Rating:** 5
**Confidence:** 3

**Summary:**

Regression losses fail to recover high-frequency details, resulting in blurred images. Meanwhile, perceptual loss introduces noise or artifacts and may even produce incorrect textures. This paper proposes the Multi-Objective Bayesian Optimization Super-Resolution (MOBOSR) framework to adjust the loss weights of both regression and perceptual losses to address their inherent conflict and combine their advantages, allowing the framework to achieve a balance between perceptual and distortion trade-off.

**Strengths:**

This is an interesting paper that applies bayesian optimization to find the Pareto frontier between perception and distortion. And the effectiveness of the method is validated through extensive experiments with SOTA SR models.

**Limitations:**

1. It's highly discouraged to use complete sentence as paper title. I suggest revising the title.
2. In Fig.1, the scatter plot on the left and the thumbnails on the right have inconsistent coordinates. For example, point c on the scatter plot is located at (30, 0.1), but on the right, it is denoted as (28.09, 0.0463). I should say Fig.1 is a bad teaser figure. Because it not only fails to clearly highlight the advantages of the proposed model but also creates unnecessary confusion.
4. Bayesian optimization should yield outputs with as high PSNR as possible and as low LPIPS as possible, but the result is that as the training epochs increase, the model tends to prioritize high PSNR while sacrificing LPIPS (from Fig.1 in main text and Fig.2 in supplementary material). Can this be explained reasonably? Or is this a possible improvement to be addressed in future work?
5. Tab.1 and Tab.2 both involve ESRGAN and the same datasets, but why are the data different? Experimental settings need to be clarified.
6. Does the MOBOSR achieve the optimal solution at (or around) 476 epochs for all datasets? If not, how do you find the optimal training epoch for new data? This needs clarification.

**Suitability:**

2

---

### Official Review · Reviewer_wEq1 · 2024-05-30

**Rating:** 5
**Confidence:** 3

**Summary:**

This paper investigates the use of different loss functions to create a pareto frontier for SISR based on a model called ESRGAN.
The work is well established although uses a strange conceptualization:
-	The use of a wrong definition for SISR models, that aim to create a higher resolution image from a lower resolution image. The paper confuses this with the typical testing model.
-	Consideres a specific metric as very special (LPIPS) instead of addopting a generic model with a large diversity of metrics.
Furthermore, the paper seems to ignore that the only reliable evaluation model is based on subjective evaluation, and not in metrics. Without a proper validation of a metric a SISR model evaluation is always somehow not reliable. Hece, considering a trained metric as a source of true quality is critical for the evaluation. A metric is only reliable after a subjective evaluation that tests it in the context it is being used.

**Strengths:**

See above

**Limitations:**

Remarks:

1 - Why the authors call LPIPS a perceptual metric? Usually, SSIM is considered a perceptual metric. Most of the developed metrics are considered perceptual. I do not understand why the authors consider LPIPS so unique for SISR.
If the authors want to use LPIPS for super-resolution evaluation need to validate its use first, typically with a subjective study. Otherwise, every is highly speculative. PSNR, on the other hand, provides a measure of the signal similarity. A PSNR below 30 is usually low quality.

2 - The formulation of eq. 2 is problematic. This is based in the principle that the high resolution image is available. However, that is not true in a real application (see remark 5 for further understanding of the problem in the 3.2 section definition of the problem.



3	- The abstract should summarize the paper. Instead, it is introductory. This is not an appropriate abstract.

4-	The authors use MOBO in the abstract without defining it.

5-	The problem formulation in 3.2 is not accurate. The paper specifies “…SISR model is to employ a neural network…”. However, we can do the same with other methodologies different from NN, whatever is the performance. Please correct.
Furthermore, this definition explains the typical methodology used in testing. However, in real applications the method intends to transform the Low-resolution image into a higher resolution. For testing purposes, the high resolution image is reduced to low resolution and the higher resolution obtained with the SISR method is evaluated compared with the high resolution.

6-The abstract should be reviewed to better reflect a summary of the paper. I could not understand what the paper is about by reading this badly formulated abstract.

**Suitability:**

2

---

### Meta-Review · Area_Chair_FvaP · 2024-07-05

**Recommendation:** Accept (Oral)
**Confidence:** 4

**Metareview:**

The work proposes a Single-image super-resolution (SISR) algorithm that uses a multi-objective optimization (MOO) method.  The method  balances perceptual quality with distortion by dynamically adjusting loss weights during training, reducing the need for manual hyperparameter tuning and lessening computational demands. The idea is novel and interesting. The paper is nicely written and the experimental validation is sufficient, showing the performance of the method both quantitatively and visually.

Reviews are in general very positive and the authors have answered most of the reviewers comments and suggestions. I think this is a good and novel contribution to ACM MM.